# Adherence to and experiences of K–12 students in modified and standard home quarantine during the SARS-CoV-2 pandemic in Missouri

**Mary Claire Worrell**[1☯*], **Sara Malone**[2☯], **Patrick Dawson**[1,3], **Stephanie A. Fritz**[2], **Ebony Thomas**[1], **Bre Peeler**[4], **Catherine Rains**[5], **Sarah C. Tinker**[1], **John C. Neatherlin**[1], **Lisa Barrios**[1], **Jon Mooney**[5], **Katie Towns**[5], **Jason Newland**[2], **Johanna S. Salzer**[1]

**1** COVID-19 Response Team, Centers for Disease Control and Prevention, Atlanta, GA, United States of America, **2** Washington University in St. Louis, St. Louis, MO, United States of America, **3** Epidemic Intelligence Service, Centers for Disease Control and Prevention, Atlanta, GA, United States of America, **4** ES Corporation, San Antonio, Texas, Atlanta, GA, United States of America, **5** Springfield-Greene County Health Department, Springfield, MO, United States of America

☯ These authors contributed equally to this work.
\* yds5@cdc.gov

**Data Availability Statement:** A limited dataset has been included as Supporting Information file 1. Qualitative data and some demographic data are not available due to ensuring privacy of

## Abstract

### Background

In November 2020, during the SARS-CoV-2 pandemic, Missouri allowed local public health jurisdictions the option to implement a modified quarantine policy allowing kindergarten through 12 (K-12) students with low-risk exposures to continue in-person learning. We assessed adherence to quarantine among participants in modified quarantine and standard home quarantine and the psychosocial impacts of quarantine on students and families.

### Methods

In January-March 2021, as part of an investigation of in-school transmission of SARS-CoV-2, parents of 586 participating K-12 students identified as a close contact with a person with SARS-CoV-2 were sent a survey to assess their activities and psychosocial impacts to the child and family.

### Results

Among the 227 (39%) survey respondents, 26 (11%) participated in modified quarantine and 201 (89%) participated in standard home quarantine. Forty-six percent of students in modified quarantine and 72% of students in standard home quarantine reported abstaining from non-school activities during quarantine. Parents of 17 (65%) students in modified quarantine and 80 (40%) in standard home quarantine reported low or neutral levels of stress in their children. Parents of students in standard home quarantine described greater stress, negative impacts to family functioning, and interruptions to educational opportunities for students.

participants. Data are available from the Washington University in St. Louis Study PI (contact co-author Jason Newland [jgnewland@wustl.edu] or Institutional PoC Cindy Terrill [terrill@wustl.edu]) for researchers who meet the criteria for access to confidential data.

**Funding:** The Missouri Department of Health and Senior Services(https://health.mo.gov/) provided funding to Washington University in St. Louis to support this study (no grant #, authors: SM, SAF, and JGN). The funders had no role in study design, data collection and analysis, decision to publish, or preparation of the manuscript.

**Competing interests:** The authors have declared that no competing interests exist.

## Conclusions

Students in modified quarantine reported lower adherence to quarantine recommendations but lower daily impact and stressors than those in standard home quarantine. Because in-school transmission of SARS-CoV-2 has been shown to be low when layered prevention strategies are in place regardless of the use of modified or standard home quarantine, this modified quarantine approach provides a reasonable option for balancing the needs of students and families with SARS-CoV-2 prevention measures.

## Background

The Coronavirus Disease 2019 (COVID-19) pandemic, caused by severe acute respiratory syndrome coronavirus 2 (SARS-CoV-2), has resulted in substantial disruptions to kindergarten through grade 12 (K-12) schools in the United States of America. As a result of the pandemic, many schools transitioned to virtual or hybrid-learning models during the 2020–2021 school year as one of many COVID-19 prevention strategies. These prevention strategies also include the implementation of universal face mask policies, physical distancing within schools, vaccination of teachers, staff, and eligible students, and the isolation and quarantine of persons who test positive with SARS-CoV-2, and individuals identified as being in close contact with them. While virtual learning has provided continued education during the pandemic, resource challenges, such as access to computers and internet, may be exacerbating existing educational inequities. Additionally, not all children are able to adapt to learning in a virtual environment [1]. Further, parents of children receiving virtual instruction more often reported their own emotional distress and report concerns about loss of work and childcare challenges [2]. Disruption to in-person learning also affects the mental health of students, especially students with existing mental health issues. A study of Italian children aged 6 to 14 in virtual learning environments found 78% experienced symptoms of anxiety and adversely impacted children's sleeping and eating habits [3]. A review of literature from the US and other countries around the world on stress related to quarantine for COVID-19 and other diseases found substantial psychological impact during and after the quarantine [4]. Loss of a school routine and reduced access to mental health services at school can worsen students' existing mental health issues [5]. Increases in mental health emergency room visits for children under 18 years of age and in emergency visits for suspected suicide attempts in female adolescents aged 12–25 years have been reported since the start of the pandemic [6, 7].

Numerous investigations have shown that the risk of classroom transmission of SARS-CoV-2 in K-12 schools is low in settings with layered prevention measures, including case investigation and contact tracing, quarantining close contacts, face mask policies, and physical distancing [8–12]. Continuation of in-person learning requires balancing the risk of transmission and health of students, teachers, and staff with the benefits of in-person education for the social, educational, and mental health needs of children and their families.

In November 2020, Missouri allowed local public health jurisdictions the option to implement a modified quarantine (MQ) policy permitting K-12 students who had classroom-associated contact with a student, teacher, or staff with COVID-19 and met masking requirements during their exposure (classified as low-risk close contacts) to continue in-person learning [13]. Under this policy, students who were in close contact with a person with COVID-19 were permitted to attend school in-person during their quarantine if the school 1) had a mask mandate, 2) classrooms were arranged to maximize physical distancing, 3) had increased hand hygiene practices, 4) screened students and staff members for COVID-19 symptoms and 5)

immediately isolated symptomatic persons. In addition, to be eligible for MQ, the exposure must be classified as low risk according to the following criteria: 1) the student was aged ≤18 years, 2) their only exposure to the person with COVID-19 was in the educational environment (e.g., a classroom), 3) they did not have prolonged (≥15 minutes) direct physical contact with the person with COVID-19, and 4) the close contact and person with COVID-19 had both been wearing masks appropriately during the time of exposure. Students in MQ were permitted to continue in-person learning, but could not attend any extracurricular activities; it was recommended that those in MQ follow the Centers for Disease Control and Prevention (CDC) standard home quarantine (SQ) recommendations [14] in existence at the time of the investigation except for attending school in-person. CDC quarantine recommendations included 1) staying home, 2) watching for fever (100.4˚F), cough, shortness of breath, or other symptoms of COVID-19, and 3) if possible, stay away from people you live with, especially people who are at higher risk for getting very sick from COVID-19. Under a SQ policy, students typically must forfeit all in-person activities including in-person instruction for 7–14 days after their last exposure. The objectives of our work were to compare the adherence to quarantine recommendations between individuals who participated in MQ and SQ, understand the psychosocial impacts of quarantine on students and their families, and understand acceptability of MQ.

## Methods

We analyzed data from a survey conducted as part of a larger investigation of secondary transmission of SARS-CoV-2 in K-12 schools. CDC, Washington University in Saint Louis, state and local health departments, and local school officials in Greene and St. Louis Counties, Missouri, conducted an investigation of COVID-19 prevention measures in K–12 public schools and their impact on in-school transmission of SARS-CoV-2 [8, 15]. During this time, schools in Greene and St. Louis Counties implemented COVID-19 mitigation strategies; however, Greene County implemented a MQ policy, while St. Louis County did not. School officials conducted contact tracing to identify school-based close contacts of students, faculty, or staff with laboratory-confirmed COVID-19. Close contact was defined "as someone who was within 6 feet of an infected person for at least 15 minutes within a 24-hour period starting from 2 days before illness onset (or, for asymptomatic cases 2 days prior to positive specimen collection) until the time the patient is isolated" [16]. Students in Greene County that do not meet the MQ eligibility completed a SQ. Parents of a child eligible for MQ could choose to not participate and keep their child home in SQ. Contacts were eligible to participate if their most recent school-based exposure was within 14 days of recruitment. Contacts were ineligible if they lived with the person with COVID-19 from the school-based exposure. We conducted a survey of the parents or guardians of school-based student close contacts to understand attitudes and practices around quarantine. This project was reviewed and approved by the Washington University in St. Louis Institutional Review Board and was conducted consistent with applicable federal law and CDC policy (see 45 C.F.R. part 46; 21 C.F.R. part 56; 42 U.S.C. §241 (d), 5 U.S.C. §552a, 44 U.S.C. §3501 et seq.). The project was deemed by the IRB to be a non-research public health surveillance activity [45 CFR 46.102(l)(2)]. Therefore, the need for informed consent was waived. However, participants provided oral agreement to participate, and parents/guardians provided oral agreement for their children aged <18 years.

### Sample

Between January 25–March 21, 2021, parents or guardians of school-based student close contacts were asked to participate in the overall investigation. In Greene County, school officials

in K-12 schools determined whether students met criteria for MQ based on contact tracing data using the criteria detailed above. Starting on March 11, the parents or guardians of student close contacts that had agreed to receive emails from the investigation and completed their quarantine (at least 14 days following the date of last exposure) were sent an online RED-Cap (version 9.5.5, Vanderbilt U) survey. For students who completed their quarantine following March 11, the parents were sent a survey 14 days after their date of last exposure. For individuals with multiple exposure events during the investigation period, only one survey was sent in the context of the most recent exposure.

### Survey questions

The survey included 11 open- or close-ended questions in English. We collected information from parents on student eligibility and decision making around MQ, psychosocial effects of quarantine on the child and parent, and non-school activities conducted during quarantine. Questions 1–7 were related to MQ and were asked only of parents of student close contacts in Greene County (Table 1). Demographic characteristics of participants were collected during the initial interview conducted as part of the larger investigation.

### Analysis

Quantitative data were managed and analyzed in R (version 3.6.3, The R Foundation). Univariate descriptive analyses were conducted to explore the responses. Statistical testing was not performed due to low sample size. For the qualitative analysis, data were analyzed using a thematic approach [17]. The dataset was initially reviewed by a coordinator (S.M.), who developed a codebook with a set of inductive codes (S1 Appendix). Codes were divided into student and parent/family categories. Two team members coded the data independently (B.P. and E.T.), and added codes as needed. A third coder reconciled any discordant codes (M.C.W.). The codes were reviewed and grouped together in themes. Initial coding agreement was 90%.

## Results

### Participants

The study team identified 586 student close contacts, 212 from Greene County and 374 from St. Louis County, that participated in the larger investigation and whose parents agreed to receive emails. Among the 586 students, 227 (39%) responded to the survey; 62 of 212 (29%) contacts from the group from Greene County and 165 of 374 (44%) from St. Louis County. Demographic characteristics for the survey participants can be found in Table 2.

### Modified quarantine decision making

Among 62 Greene County survey respondents, 35 (57%) close contacts were eligible for MQ, and 26 (43%) participated in MQ. All close contacts (165) from St. Louis County participated in SQ and 36 contacts from Greene County participated in SQ for a total of 201 participants in SQ (Table 2). The most common reason (n = 24, 96%) for the decision to participate in MQ was following the school's recommendation, followed by parents not thinking that their child continuing in-person education would put their child's health at risk (n = 16, 62%). Reasons that MQ eligible students did not participate in MQ were that the quarantine coincided with a school break or dismissals due to a snowstorm (n = 4, 44%), followed by concerns for health of the child and concerns for safety of other students/staff at school (n = 3, 33%) or in community (n = 3, 33%). Among the 27 Greene County students who were not eligible for MQ, the parents of 17 (63%) would have accepted MQ if it had been offered. Among the 9 parents who would

**Table 1. Survey questions and response options for parents/guardians of students participating in modified and standard quarantine.**

| # | Question | Options | Branching Logic | Site Asked |
|---|----------|---------|-----------------|------------|
| Q1 | During your child's recent quarantine period, was your child allowed to continue in-person learning at school under the modified quarantine policy? | Yes<br>No | | Greene County |
| Q2 | If your child had been offered to participate in modified quarantine, would you have allowed your child to attend in-person learning during their quarantine period? | Yes<br>No<br>Unsure | If no to Q1 | Greene County |
| Q2a | Why did you select no or unsure? | Open-ended | If selected "No" or Unsure from Q2 | Greene County |
| Q3 | Did your child attend school for in-person learning during any part of their modified quarantine period? | Yes<br>No | If yes to Q1 | Greene County |
| Q4 | What were the reasons you did NOT allow your child to participate? [select all that apply] | Concern for health of your child<br>Concern for health of other family members<br>Concern for safety for other students/staff at school or in community<br>Virtual learning was preferred or easier for student or family<br>Availability of childcare options (e.g., parent/guardian or other family member home or paid childcare was a preferred option)<br>Did not understand the modified quarantine policy<br>Child stayed home because their classmates stayed home<br>Received advice to not participate in modified quarantine from a family member, friend, or healthcare professional<br>Other<br>Prefer not to say | If no to Q3 | Greene County |
| Q4a | What were the other reasons you did not allow your child to participate? | Open-ended | If selected "other" in Q4 | Greene County |
| Q5 | What were the reasons you ALLOWED your child to participate? (Select all that apply) | Followed school's recommendation<br>Did not think your child continuing to attend school would put their health at any greater risk<br>Did not think your child continuing to attend school would put any other family members health at greater risk<br>Did not think your child continuing to attend school would affect the safety of the students/staff at school or in community<br>Lack of availability of virtual learning if child was out of in-person learning<br>Worried that staying home would harm your child's mental health<br>Challenges associated with virtual learning<br>Lack of childcare options (e.g., parent/guardian or other family member home or paid childcare)<br>Followed what other parents/guardians of classmates decided to do<br>Received advice to participate in modified quarantine from a family member, friend, or healthcare professional<br>Prefer in-person learning<br>Other<br>Prefer not to say | | Greene County |
| Q5a | What were the other reasons you allowed your child to participate? | Open-ended | If selected "Other" in Q6 | Greene County |
| Q6 | Do you think your child attending in-person learning during their quarantine poses a risk to the health of teachers or other staff members at school? | Yes<br>No<br>Prefer not to say | If selected | Greene County |

(*Continued*)

**Table 1.** (Continued)

| # | Question | Options | Branching Logic | Site Asked |
|---|---|---|---|---|
| Q7 | Would you feel safe having your child in the classroom with other students who are allowed to attend in-person learning during their quarantine period (i.e., they had been in close contact with a person known to have COVID-19 at school but both had worn masks)? | Yes<br>No<br>Prefer not to say | | Greene County |
| Q8 | How much was your family's day-to-day life impacted by your child's quarantine period? | Strongly negatively impacted<br>Somewhat negatively impacted<br>Neither negatively nor positively impacted<br>Somewhat positively impacted<br>Strongly positively impacted<br>Prefer not to say | | Greene County and St. Louis County |
| Q9 | How stressful was your child's day-to-day life during their quarantine period? | Much more stressful than usual<br>Somewhat more stressful than usual<br>Neither more nor less stressful than usual<br>Somewhat less stressful than usual<br>Much less stressful than usual<br>Prefer not to say | | Greene County and St. Louis County |
| Q10 | Please share any details about how you and your family were most affected by quarantine. | Open-ended | | Greene County and St. Louis County |
| Q11 | During your child's quarantine period, did your child do any of the following activities outside of in-person learning? (Select all that apply) | Interact in person with classmates who were also quarantined<br>Interact in person with non-quarantined friends or classmates from their school<br>Interact in person with non-quarantined friends not from their school<br>Interact with family members who do not live in your household<br>Go to a restaurant to dine in<br>Attend events (e.g., church, parties, movies, entertainment, etc.)<br>Enter stores or businesses (e.g., grocery shopping, shopping, takeout food, etc.)<br>Go to work or volunteer<br>Participate in afterschool or extracurricular activities (e.g., sports, band, dance, etc.)<br>Travel outside of your city<br>Cancel social events or choose not to participate in planned activities (e.g., church, parties, etc.)<br>Leave home for reasons other than those mentioned above<br>Prefer not to say<br>Other activities not asked in above questions you would like to share | | Greene County and St. Louis County |
| Q11a | What other activities did you participate in during your (your child's) quarantine period? | Open-ended | If selected "Other" in Q11 | Greene County and St. Louis County |

not have accepted MQ or were unsure, the main reason was concerns of exposing others (n = 5, 63%). One parent did not answer the question. Several parents noted that their decision would depend on the nature of exposure (i.e. time, location, masking).

## Quarantine behaviors

Overall, 12 (46%) parents of students in MQ reported the student refrained from participating in non-school-associated activities during the quarantine period while 145 (72%) parents of students in SQ reported refraining from non-school-associated activities. Parents of SQ

**Table 2. Demographic characteristics of students in close contact with persons who tested positive for SARS-CoV-2.**

| Student Characteristics | All, n (%) | Greene County Modified Quarantine, n(%) | Greene County Standard Quarantine, n(%) | St. Louis County Standard Quarantine, n(%) | Total Standard Quarantine, n(%) |
|---|---|---|---|---|---|
| **Total** | 227 | 26 | 36 | 165 | 201 |
| **Race** | | | | | |
| White | 185 (82) | 22 (85) | 29 (81) | 134 (81) | 163 (80) |
| Other | 42 (19) | 4 (15) | 7 (19) | 31 (19) | 38 (20) |
| **Ethnicity** | | | | | |
| Hispanic/Latino | 7 (3) | 0 (0) | 1 (3) | 6 (4) | 7 (0) |
| Non-Hispanic/Latino | 218 (96) | 26 (100) | 34 (94) | 158 (96) | 192 (100) |
| Unknown/Prefer not to answer | 2 (1) | 0 (0) | 1 (3) | 1 (1) | 2 (0) |
| **School Grade Level** | | | | | |
| Elementary school grade (grades K-5) | 62 (27) | 6 (23) | 16 (44) | 40 (24) | 56 (30) |
| Middle school grade (grades 6–8) | 96 (42) | 3 (12) | 15 (42) | 78 (47) | 93 (50) |
| High school grade (grades 9–12) | 69 (30) | 17 (65) | 5 (14) | 47 (29) | 52 (30) |

Abbreviations: K–12 = kindergarten through grade 12; K-5 = kindergarten through grade 5
Gender was not presented due to small cell size. Overall, students were 51% female (54% in MQ, 39% in SQ in Greene County, and 53% in SQ in St. Louis County). One person reported other gender, which included transgender, non-binary, or other gender.

students reported similar frequencies of activities between the two counties. Nine (35%) MQ parents and 19 (9%) parents of SQ students reported their child had interactions with other students outside of school. Among those in quarantine that did not interact with other students outside of school, 12 (67%) MQ students and 152 (84%) SQ students reported not participating in other activities. Students who did interact with other students outside of school refrained from participating in other activities during their quarantine at lower frequencies, two (25%) MQ students and seven (37%) SQ students (Table 3).

Interaction with other quarantined or non-quarantined friends was the most common non-school-associated activity for students in MQ (35%) and SQ (9%). Thirty-five percent of students in MQ, 33% of students in SQ in Greene County, and 50% of students in St. Louis County in SQ reported canceling social events or not participating in planned non-school-associated activities (Table 4).

## Quarantine psychosocial impacts

Parents provided data on how quarantine affected the family's day-to-day life and their child's overall stress. The parents of 10 (38%) students in MQ, 19 (53%) Greene County students in SQ, and 111 (67%) St. Louis County students in SQ reported negative impacts of quarantine on the family's day-to-day life. When asked about the child's stress during their quarantine, the parents of 9 (35%) students in MQ (Fig 1A), 19 (53%) Greene County students in SQ (Fig 1B) and 102 (62%) St. Louis County students in SQ reported more levels of stress (Fig 1C).

One-hundred and eighteen parents (51%) supplied expanded qualitative information about the child's quarantine and its impact on the family (Table 1 - Q10). For responses that included information about the child's experiences, four themes emerged out of the codes: (1) negative mental health impacts, (2) activity disruptions, (3) educational impacts, and (4) positive

**Table 3. Summary of non-school related activities reported by parents/guardian of K-12 student close contacts.**

| Summary of reported activities conducted during the quarantine period | All, n(%) | Modified Quarantine, n(%) | Standard Quarantine, n(%) |
|---|---|---|---|
| Total | 227 | 26 | 201 |
| All activities[a] | | | |
| 0 activities | 157 (69.2) | 12 (46.2) | 145 (72.1) |
| 1–2 activities | 48 (21.1) | 9 (34.6) | 39 (19.4) |
| 3 or more activities | 20 (8.8) | 4 (15.4) | 16 (8.0) |
| Prefer not to answer | 2 (0.9) | 1 (3.8) | 0 (0) |
| Interacted with other students outside of school[c] | 28 (12.3) | 9 (34.6) | 19 (9.4) |
| Additional non-school activities[b] | | | |
| 0 activities | 9 (32.1) | 2 (22.2) | 7 (36.8) |
| 1–2 activities | 9 (32.1) | 4 (44.4) | 5 (26.3) |
| 3 or more activities | 10 (35.7) | 3 (33.3) | 7 (36.8) |
| Prefer not to answer | 0 (0) | 0 (0) | 0 (0) |
| Did not interact with other students outside of school | 200 (88.1) | 18 (69.2) | 182 (90.5) |
| Additional non-school activities | | | |
| 0 activities | 164 (82) | 12 (66.7) | 152 (83.5) |
| 1–2 activities | 29 (14.5) | 4 (22.2) | 5 (13.7) |
| 3 or more activities | 5 (2.5) | 1 (5.6) | 4 (2.2) |
| Prefer not to answer | 2 (1) | 1 (5.6) | 1 (0.5) |

a: Includes both interactions with students and other activities

b: Other activities include: interacting with non-school friends, interacting with family members outside their household, going to a restaurant to dine in, attend events (such as church, parties, movies, etc), enter stores or businesses (such as grocery shopping, shopping, takeout food, etc), go to work or volunteer, visit gym or play sports, travel outside the city, or leave home for other reasons.

c: Includes interaction with students outside of school who are quarantined and not quarantined

experiences. Examples of quotes, arranged by quarantine type and stress level, can be found in Fig 1. Parents of students in SQ in St. Louis County more frequently reported negative mental health impacts (n = 46, 28%) than parents of students in MQ (n = 4, 15%) and parents of

**Table 4. Non-school related activities reported by parents/guardian of K-12 student close contacts.**

| Activities | All, n(%) | Modified Quarantine, n(%) | Standard Quarantine, n(%) |
|---|---|---|---|
| Total | 227 | 26 | 201 |
| Interact in person with other quarantined students | 12 (5.3) | 5 (19) | 7 (3.5) |
| Interact in person with non-quarantined friends from your school | 22 (9.7) | 6 (23) | 16 (8) |
| Interact in person with non-quarantined friends not from your school | 18 (7.9) | 4 (15) | 14 (7) |
| Interact with family members who do not live in your household | 14 (6.2) | 2 (8) | 12 (6) |
| Go to a restaurant to dine in | 14 (6.2) | 2 (8) | 12 (6) |
| Attend events (e.g., church, parties, movies, entertainment, etc.) | 10 (4.4) | 2 (8) | 8 (4) |
| Enter stores or businesses (e.g., grocery shopping, shopping, takeout food, etc.) | 22 (9.7) | 3 (12) | 19 (9.5) |
| Go to work or volunteer | 4 (1.8) | 0 (0) | 4 (2) |
| Visit gym or play sports | 10 (4.4) | 3 (12) | 7 (3.5) |
| Travel outside of your city | 9 (4) | 3 (12) | 6 (3) |
| Cancel social events or choose not to participate in planned activities (e.g., church, parties, etc.) | 109 (48) | 9 (35) | 100 (49.8) |
| Leave home for reasons other than those mentioned above | 13 (5.7) | 3 (12) | 10 (5) |
| Other activities | 12 (5.3) | 0 (0) | 12 (6) |
| Prefer not to say | 2 (0.9) | 1 (4) | 1 (0.5) |

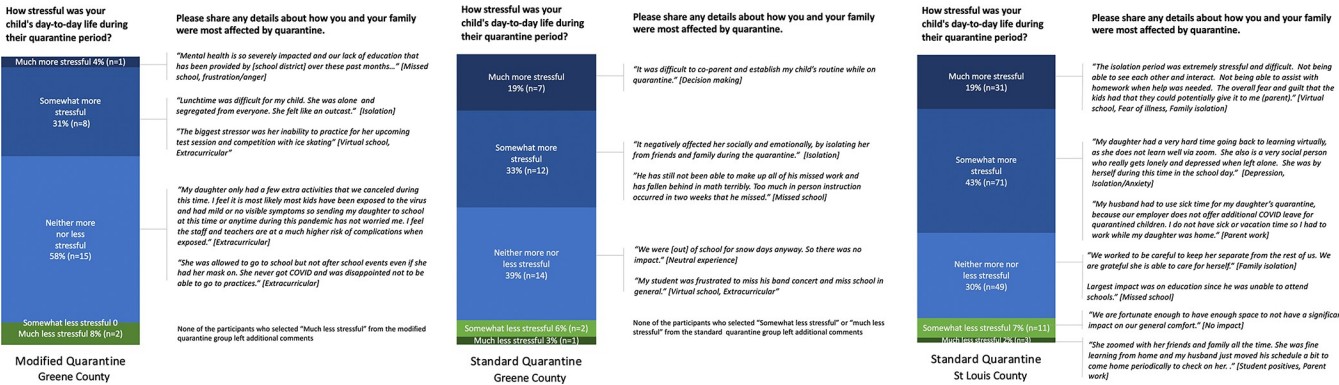

**Fig 1.** Reported quarantine experiences from parents/guardians of modified quarantine (a) and standard quarantine (b) in Greene County, Missouri and in standard quarantine (c) in St. Louis, County Missouri, January-March 2021.

students in SQ in Greene County (n = 2, 6%). Parents of both SQ and MQ students in both counties described students as having an array of mental health impacts, including increased social isolation, anxiety, and frustration. Additionally, parents of students in SQ in St. Louis County described the psychosocial impact on children including what they described as depression (n = 10, 6%), which was not reported by MQ parents. Not being able to participate in extracurricular activities was the most frequent concern reported by parents of MQ students (n = 5, 19%) furthering these feelings of social isolation and frustration. For students in SQ, activity impacts (Greene County n = 2, 6%; St. Louis County n = 26, 16%) as well as educational impacts (Greene County n = 12, 33%; St. Louis County n = 52, 32%) were reported. There were descriptions of students in SQ missing the benefits of in-person learning as well as having difficulties, both logistical and psychosocial, with virtual learning. No educational impacts were reported by the parents with children in MQ. Some parents reported neutral or positives impacts of SQ, such as reduced commute times and increased flexibility for a child's schedule.

In the qualitative analysis, themes emerged regarding the consequences of student quarantines on parents. Three parental areas were identified: (1) challenges with work, (2) decision making around safety for student and family, and (3) positives of quarantine. Parents of students in SQ described disruption of their daily routine, including missing work and having to aid with virtual schooling (Greene County, n = 7, 20%; St. Louis County, n = 17, 10%), while no MQ parents reported these disruptions. Parents of children in MQ and SQ also described the stress of having to make decisions about splitting the family within the home and managing the student's stress around being ill or getting other family members sick, with nearly equal frequency (4–6%). Finally, some parents mentioned reductions in stress due to having fewer commitments during quarantine (3%).

## Discussion

Prior literature suggests that students in virtual learning and standard home quarantine experience associated stress and anxiety. MQ is a new strategy being employed in parts of the state of Missouri, but limited data are available regarding the acceptability of the MQ strategy. This work aimed to explore parental reports of students' behaviors and attitudes of those in MQ versus SQ being used by most of the schools around the country. We first explored the acceptance of MQ as an alternative strategy to standard home quarantine. MQ was accepted by most parent respondents of eligible students, but for some parents the decision may depend

on the type of exposure. There may have been additional participation in MQ if the quarantine period had not coincided with school breaks and school cancellations due to weather, which was the most frequent reason for not participating. Parents most often reported using the school's advice as part of their decision making, demonstrating a high level of trust with school guidance. Some parents noted concerns regarding their child's health and concerns for safety for other students and staff at school or in the community for those who were in MQ. MQ provides parents the opportunity to decide whether keeping their child home during quarantine or allow them to attend school is the best option for their family.

In allowing students to attend school during quarantine, there was concern that this strategy could decrease adherence to quarantine in other aspects of a student's life. Therefore, we surveyed parents about their child's activities to understand if MQ students showed lower adherence to quarantine guidelines (i.e. increased participation in non-school-associated activities), than those in SQ. Parents of students in MQ reported participating in more non-school-associated activities during their quarantine in comparison to those in SQ. However, many parents reported these interactions to be with other students (quarantined and non-quarantined) from school signaling that keeping children in schools during quarantine could increase interaction with schoolmates outside of the school setting. Among MQ students who did not interact with other schoolmates, there were lower frequencies of reported other non-school-associated activities. Similar trends can be seen in the SQ students, suggesting that if students interact with other students during their quarantine, they are more likely to participate in other activities as well. While students in MQ reportedly conducted more non-school associated activities than those in SQ, the risk of secondary transmission was found to be low in the larger investigation [unpublished data]. Schools should continue to provide strong messaging about quarantine recommendations for close contacts in both MQ and SQ as 17% of all parents reported non-school-associated activities during their quarantine.

While prior studies reported high rates of stress amongst students [3, 5, 6], our work explored the potential impact of MQ strategies on students' stress. Parents of students in SQ reported higher frequencies of negative impacts on the family's day-to-day life and increased stress of students in SQ when compared to those in MQ. Parents of students in MQ and SQ described how the quarantine of students also affected both the child in terms of mental health and education and caused disruptions to parents' daily routines, household dynamics, and work. This work highlights the instances of stress, negative impact to family functioning, and frustration with reduced educational opportunity for students that complement findings from other interviews with children during the COVID pandemic [3].

Virtual school was a major concern for parents of students in SQ and is an important example of the wide-ranging effects of quarantine on students. Virtual school affected the student themselves, whose parents reported struggling with virtual learning and impacting education, as well as larger effects on parents and the family, which had to contend with monitoring virtual school that impacted the parents' work. While there is limited evidence about this topic, this investigation begins to explore how virtual school not only affected the child's education, but also affected their emotional well-being in terms of anxiety and feelings of isolation and created a challenge for parents trying to work and support their child with schooling. This work expands on prior studies through a comparison of stress between different quarantine types. While mental health effects were more frequently related in SQ, students in MQ were also impacted and focused on missing extracurricular activities and experiences of isolation and anxiety. For example, some parents noted that students felt isolated because the school had separate seating for MQ students at lunch. This suggests that areas of intervention for those in MQ may also need to be developed.

Finally, this survey captured some parental stress around quarantine and its impact on the family unit. While parents with children in MQ reported fewer parental and familial impacts in comparison to parents with students in SQ, this work aligns with other reports of family stress during the COVID-19 pandemic [18]. The daily impacts on the family are likely different depending on the family's situation, such as work flexibility and ability to work from home.

The findings of this report are subject to several limitations. First, with a 32% response rate, we may not have captured the full spectrum of views and experiences of parents of all close contacts, and we were unable to perform statistical testing due to insufficient power. Second, some parents received the survey several weeks following their child's quarantine and parents may have had a different perception of their experience as time passed. Third, for the reporting of activities completed during a quarantine, social desirability bias could have skewed responses toward fewer reported activities conducted during quarantine. Additionally, we collected binary data on whether an activity was conducted during the quarantine period, but we did not collect quantitative data on how many times that activity was conducted. Therefore, the results only can report whether certain activities were conducted and not the extent to which student close contacts did those activities. Fourth, a snowstorm affecting both Greene and St. Louis Counties during the investigation led to school cancellations and several school breaks affected the quarantine of some students. These students and families may have had different experiences than students who were quarantined during normal school session. Lastly, our investigation could be affected by extreme bias. Respondents with strong views on quarantine could have been more likely to complete the survey which could bias our results, particularly negative extremes. For the qualitative analysis, participants with negative experiences were more likely to provide additional information compared to the participants with neutral or positive experiences, which could bias our data towards more negative impact themes.

## Conclusion

Findings suggest that the negative impacts on an individual's daily life and stress on student close contacts and their families may occur less frequently in students in MQ than those in SQ and that the MQ policy has been accepted by many parents in Greene County as an alternative to SQ. However, more rigorous investigation is needed to understand the impacts of different quarantine policies on students and their families. Despite students in MQ having increased frequencies of participation in non-school associated activities as compared to students in SQ, the risk of transmission of SARS-CoV-2 among students in MQ has been demonstrated to be low. Thus, the MQ approach taken by Greene County in schools implementing layered prevention strategies, such as vaccination, physical distancing, and masking of unvaccinated individuals, may provide an option for balancing mitigation of SARS-CoV-2 transmission, maintaining in-person education, and decreasing the negative psychosocial impacts on student close contacts and their families. It is important to increase education around the quarantine recommendations for both students in MQ and SQ to reduce frequency of non-school activities during quarantine.

## Supporting information

**S1 File. Limited dataset.**
(XLSX)

**S1 Appendix. Coding structure for qualitative thematic analysis of parent responses to the open-ended question.**
(DOCX)

## Acknowledgments

All the students, families, educators, nurses, administrators, and staff members from participating schools and school districts in Saint Louis and Springfield, Missouri; CDC COVID-19 Response Team (Catherine Rasberry, Elizabeth Haller, Hailey Reid, Monica LaBelle, Neha Cramer); Saint Louis University (Adrienne Beckett-Ansa, Allie Bodin, Ashley Gomel, Rachel Leimkuehler, Elena Dalleo Locascio, Mary Beal, Rachel L. Mazzara, Riley Voss, Ruband Mahmood, Samantha Hayes); Springfield-Greene County Health Department (Kathryn Wall, Brad Stulce, Sean Barnhill); University of Missouri School of Medicine (Dr. David Haustein, Evan Garrad, Hosea Covington, Tricia Haynes, Spencer Blake Price), and Washington University in St. Louis (Alex Plattner, Cole Tipton, Ian T. Lackey, Jenna Rideout, Savanah Low, Suong Nguyen).

**Disclaimer:** The findings and conclusions in this report are those of the authors and do not necessarily represent the official position of the Centers for Disease Control and Prevention.

## Author Contributions

**Conceptualization:** Mary Claire Worrell, Sara Malone, Patrick Dawson, Stephanie A. Fritz, Ebony Thomas, Catherine Rains, Sarah C. Tinker, John C. Neatherlin, Lisa Barrios, Jon Mooney, Katie Towns, Jason Newland, Johanna S. Salzer.

**Data curation:** Mary Claire Worrell, Sara Malone, Patrick Dawson.

**Formal analysis:** Mary Claire Worrell, Sara Malone, Ebony Thomas, Bre Peeler.

**Funding acquisition:** Jason Newland.

**Investigation:** Mary Claire Worrell, Sara Malone, Patrick Dawson, Stephanie A. Fritz, Ebony Thomas, Catherine Rains, Sarah C. Tinker, John C. Neatherlin, Lisa Barrios, Jon Mooney, Katie Towns, Jason Newland, Johanna S. Salzer.

**Methodology:** Mary Claire Worrell, Sara Malone, Patrick Dawson, Stephanie A. Fritz, Catherine Rains, Sarah C. Tinker, John C. Neatherlin, Lisa Barrios, Jon Mooney, Katie Towns, Jason Newland, Johanna S. Salzer.

**Project administration:** Mary Claire Worrell, Sara Malone, Patrick Dawson, Stephanie A. Fritz, Catherine Rains, Sarah C. Tinker, John C. Neatherlin, Lisa Barrios, Jon Mooney, Katie Towns, Jason Newland, Johanna S. Salzer.

**Resources:** Sara Malone, Jon Mooney, Katie Towns, Jason Newland.

**Software:** Sara Malone, Jason Newland.

**Supervision:** Mary Claire Worrell, Sara Malone, Patrick Dawson, Stephanie A. Fritz, Sarah C. Tinker, Lisa Barrios, Jason Newland, Johanna S. Salzer.

**Validation:** Mary Claire Worrell, Sara Malone, Patrick Dawson, Stephanie A. Fritz, Bre Peeler, Sarah C. Tinker, Jason Newland, Johanna S. Salzer.

**Visualization:** Mary Claire Worrell, Sara Malone.

**Writing – original draft:** Mary Claire Worrell, Sara Malone.

**Writing – review & editing:** Mary Claire Worrell, Sara Malone, Patrick Dawson, Stephanie A. Fritz, Ebony Thomas, Bre Peeler, Catherine Rains, Sarah C. Tinker, John C. Neatherlin, Lisa Barrios, Jon Mooney, Katie Towns, Jason Newland, Johanna S. Salzer.

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
