## [Decision Letter · Decision Letter 0]

5 Oct 2022

PONE-D-22-25634Adherence to and experiences of K–12 students in modified and standard home quarantine during the SARS-CoV-2 pandemic in MissouriPLOS ONE

Dear Dr. Worrell,

Thank you for submitting your manuscript to PLOS ONE. After careful consideration, we feel that it has merit but does not fully meet PLOS ONE’s publication criteria as it currently stands. Therefore, we invite you to submit a revised version of the manuscript that addresses the points raised during the review process.

We look forward to receiving your revised manuscript.

Kind regards,

Alejandro Vega-Muñoz, Ph.D.

Academic Editor

PLOS ONE

“All the students, families, educators, nurses, administrators, and staff members from participating schools and school districts in Saint Louis and Springfield, Missouri; CDC COVID-19 Response Team (Catherine Rasberry, Elizabeth Haller, Hailey Reid, Monica LaBelle, Neha Cramer); Saint Louis University (Adrienne Beckett-Ansa, Allie Bodin, Ashley Gomel, Rachel Leimkuehler, Elena Dalleo Locascio, Mary Beal, Rachel L. Mazzara, Riley Voss, Ruband Mahmood, Samantha Hayes); Springfield-Greene County Health Department (Kathryn Wall, Brad Stulce, Sean Barnhill); University of Missouri School of Medicine (Dr. David Haustein, Evan Garrad, Hosea Covington, Tricia Haynes, Spencer Blake Price), and Washington University in St. Louis (Alex Plattner, Cole Tipton, Ian T. Lackey, Jenna Rideout, Savanah Low, Suong Nguyen). We thank the Missouri Department of Health and Senior Services who funded this investigation.

“The Missouri Department of Health and Senior Services(https://health.mo.gov/)

provided funding to Washington University in St. Louis to support this study (no grant #,

authors: SM, SAF, and JGN). The funders had no role in study design, data collection and analysis, decision to publish, or preparation of the

manuscript.

6. Thank you for stating the following in your Competing Interests section: 

“The authors have declared that no competing interests exist.

Powered by”

 This information should be included in your cover letter; we will change the online submission form on your behalf."

Reviewers' comments:

Reviewer's Responses to Questions

**Comments to the Author**

1. Is the manuscript technically sound, and do the data support the conclusions?

Reviewer #1: Yes

Reviewer #2: Partly

Reviewer #3: Yes

Reviewer #4: Yes

2. Has the statistical analysis been performed appropriately and rigorously? 

Reviewer #1: Yes

Reviewer #2: Yes

Reviewer #3: Yes

Reviewer #4: Yes

3. Have the authors made all data underlying the findings in their manuscript fully available?

Reviewer #1: Yes

Reviewer #2: Yes

Reviewer #3: Yes

Reviewer #4: Yes

4. Is the manuscript presented in an intelligible fashion and written in standard English?

Reviewer #1: Yes

Reviewer #2: Yes

Reviewer #3: Yes

Reviewer #4: Yes

5. Review Comments to the Author

Reviewer #1: Congrats for this excellent work. The study contributions are remarkable, and the manuscript has been very well-written and structured. The conclusions are well supported by the results. No recommendations are provided and it mau be accepted in its current form.

Reviewer #2: In my opinion, the study is very interesting and worth being published. Nevertheless, the sections are not sufficiently developed to exhibit the value of the research undertaken by the authors. For this reason, I feel I can give the following suggestions:

Literature review

I suggest that the authors provide structure to their study. Additionally, the literature review needs to be properly structured (ie. explanation of the main theories, the different theoretical positions and contestations within the field and others).

Methodology

The authors should provide detailed information on the research design, the population of the study, the sampling techniques used and adequate information on the measures for the instrument.

Please summarize the main findings of the study.

It is very difficult for me to identify the main findings of the study. I have suggested that the authors provide very clear findings of their study.

Please highlight the limitations and strengths.

The current articles lacks structure and it is very difficult to read and understand. The authors should provide clear findings of their study. In its current form, readers will struggle to identify what the findings of the current study are and how the findings address weaknesses in previous studies.

I think the authors can easily follow the suggestions I have given in this review and make a new version of their interesting paper.

All best wishes.

Reviewer #3: The manuscript was well written and organized. It reported the results of MQ and SQ regarding psychosocial health using percentages. Although no statistical testing was used, it still provided useful information. My only suggestion is to tone down the implications of the findings a bit regarding MQ strategies since the sample size was small, as the authors acknowledged.

Reviewer #4: The manuscript title: Adherence to and experiences of K–12 students in modified and standard home quarantine during the SARS-CoV-2 pandemic in Missouri., Overall, this manuscript is quite well written., It appropriate to publish to journal. Please recheck all references, table and figure again.

6. PLOS authors have the option to publish the peer review history of their article (what does this mean?). If published, this will include your full peer review and any attached files.

Reviewer #1: No

Reviewer #2: **Yes: **Teresa Pozo-Rico, PhD (Educational Psychology)

Reviewer #3: No

Reviewer #4: No

---

## [Author Response · Author response to Decision Letter 0]

26 Oct 2022

Thank you for your feedback on our manuscript. We have provided responses to your comments below.

Reviewers' comments:

Comments to the Author

1. Is the manuscript technically sound, and do the data support the conclusions?

Reviewer #1: Yes

Reviewer #2: Partly

Reviewer #3: Yes

Reviewer #4: Yes

2. Has the statistical analysis been performed appropriately and rigorously?

Reviewer #1: Yes

Reviewer #2: Yes

Reviewer #3: Yes

Reviewer #4: Yes

3. Have the authors made all data underlying the findings in their manuscript fully available?

Reviewer #1: Yes

Reviewer #2: Yes

Reviewer #3: Yes

Reviewer #4: Yes

4. Is the manuscript presented in an intelligible fashion and written in standard English?

Reviewer #1: Yes

Reviewer #2: Yes

Reviewer #3: Yes

Reviewer #4: Yes

5. Review Comments to the Author

Reviewer #1: Congrats for this excellent work. The study contributions are remarkable, and the manuscript has been very well-written and structured. The conclusions are well supported by the results. No recommendations are provided and it mau be accepted in its current form.

• Thank you for your review!

Reviewer #2: In my opinion, the study is very interesting and worth being published. Nevertheless, the sections are not sufficiently developed to exhibit the value of the research undertaken by the authors. For this reason, I feel I can give the following suggestions:

• Thank you for your review and suggestions.

Literature review

I suggest that the authors provide structure to their study. Additionally, the literature review needs to be properly structured (ie. explanation of the main theories, the different theoretical positions and contestations within the field and others).

• On lines 67-75, we discussed some of the previous literature regarding mental health impacts of COVID-19 on K-12 students; however, there are not investigations that compared different quarantine policies on these mental health impacts. On lines 76-81, we discussed previous literature on COVID-19 transmission in the K-12 setting.

Methodology

The authors should provide detailed information on the research design, the population of the study, the sampling techniques used and adequate information on the measures for the instrument.

• We conducted a survey as part of a larger investigation into secondary transmission of SARS-CoV-2 in K-12 schools. The methods for the larger investigation can be found in a manuscript that has recently been accepted with PlosOne (lines 108-112)

• The population of the study is students in modified quarantine and standard quarantine in Missouri (lines 112-121, 128-137), where we defined the criteria for close contact and the two types of quarantine.

• The measures for the instrument were detailed in lines 140-145 and in Table 1, where you can find the questions asked on the survey.

Please summarize the main findings of the study.

It is very difficult for me to identify the main findings of the study. I have suggested that the authors provide very clear findings of their study.

• We discuss the findings in the Discussion section and then summarize the main findings in the conclusion “Findings suggest that the negative impacts of on an individual’s daily life and stress on student close contacts and their families may occur less frequently in students in MQ than those in SQ and that the MQ policy has been accepted by many parents in Greene County as an alternative to SQ. Despite students in MQ having increased frequencies of participation in non-school associated activities as compared to students in SQ, the risk of transmission of SARS-CoV-2 among students in MQ has been demonstrated to be low.” (Lines 350-365)

Please highlight the limitations and strengths.

• We discussed the limitations of the investigation on lines 340-358. 

The current articles lacks structure and it is very difficult to read and understand. The authors should provide clear findings of their study. In its current form, readers will struggle to identify what the findings of the current study are and how the findings address weaknesses in previous studies.

I think the authors can easily follow the suggestions I have given in this review and make a new version of their interesting paper.

All best wishes.

• Thank you for your suggestions. We had included section sub-headings to help guide the reader through the article.

Reviewer #3: The manuscript was well written and organized. It reported the results of MQ and SQ regarding psychosocial health using percentages. Although no statistical testing was used, it still provided useful information. My only suggestion is to tone down the implications of the findings a bit regarding MQ strategies since the sample size was small, as the authors acknowledged.

• Thank you for your review. We have added an additional statement in the conclusion to tone down the implications as suggested and note that more robust investigations are needed in this area. 

Reviewer #4: The manuscript title: Adherence to and experiences of K–12 students in modified and standard home quarantine during the SARS-CoV-2 pandemic in Missouri., Overall, this manuscript is quite well written., It appropriate to publish to journal. Please recheck all references, table and figure again.

• Thank you for your review. We check the references are removed one duplicate reference.

---

## [Editor Report · Decision Letter 1]

21 Nov 2022

Adherence to and experiences of K–12 students in modified and standard home quarantine during the SARS-CoV-2 pandemic in Missouri

PONE-D-22-25634R1

Dear Dr. Worrell,

We’re pleased to inform you that your manuscript has been judged scientifically suitable for publication and will be formally accepted for publication once it meets all outstanding technical requirements.

Kind regards,

Alejandro Vega-Muñoz, Ph.D.

Academic Editor

PLOS ONE
---

## [Editor Report · Acceptance letter]

29 Nov 2022

PONE-D-22-25634R1 

Adherence to and experiences of K–12 students in modified and standard home quarantine during the SARS-CoV-2 pandemic in Missouri 

Dear Dr. Worrell:

I'm pleased to inform you that your manuscript has been deemed suitable for publication in PLOS ONE. Congratulations! Your manuscript is now with our production department. 

Kind regards, 

on behalf of

Dr. Alejandro Vega-Muñoz 

Academic Editor

PLOS ONE